# Time to treatment disruption in children with HIV-1 randomized to initial antiretroviral therapy with protease inhibitors versus non-nucleoside reverse transcriptase inhibitors

**Dwight E. Yin** [1,2,3¤]*, **Christina Ludema** [4], **Stephen R. Cole** [3], **Carol E. Golin** [5,6], **William C. Miller** [7], **Meredith G. Warshaw** [8], **Ross E. McKinney, Jr.** [9¶], **on behalf of the PENPACT-1 (PENTA 9 / PACTG 390) Study Team** [¶]

1 Division of Infectious Diseases and Division of Clinical Pharmacology, Toxicology and Therapeutic Innovation, Department of Pediatrics, Children's Mercy-Kansas City and University of Missouri-Kansas City, Kansas City, Missouri, United States of America, 2 Division of Infectious Diseases, Department of Pediatrics, Duke University Medical Center, Durham, North Carolina, United States of America, 3 Department of Epidemiology, Gillings School of Global Public Health, University of North Carolina at Chapel Hill, Chapel Hill, North Carolina, United States of America, 4 Department of Epidemiology and Biostatistics, School of Public Health, Indiana University, Bloomington, Indiana, United States of America, 5 Department of Health Behavior, Gillings School of Global Public Health, University of North Carolina at Chapel Hill, Chapel Hill, North Carolina, United States of America, 6 Department of Medicine, School of Medicine, University of North Carolina at Chapel Hill, Chapel Hill, North Carolina, United States of America, 7 Department of Epidemiology, College of Public Health, The Ohio State University, Columbus, Ohio, United States of America, 8 Center for Biostatistics in AIDS Research, Harvard T.H. Chan School of Public Health, Boston, Massachusetts, United States of America, 9 Association of American Medical Colleges, District of Columbia, Washington, United States of America

¤ Current address: Division of AIDS, National Institute of Allergy and Infectious Diseases, National Institutes of Health, Rockville, Maryland, United States of America (Note: This research was conducted while employed with Children's Mercy-Kansas City.)
¶ Membership of the PENPACT-1 (PENTA 9 / PACTG 390) Study Team is provided in the Acknowledgments.
* dwight_yin@yahoo.com

**Data Availability Statement:** These data were obtained by a data request by the authors and are under the management of IMPAACT and PENTA,

## Abstract

### Background

Choice of initial antiretroviral therapy regimen may help children with HIV maintain optimal, continuous therapy. We assessed treatment-naïve children for differences in time to treatment disruption across randomly-assigned protease inhibitor versus non-nucleoside reverse transcriptase inhibitor-based initial antiretroviral therapy.

### Methods

We performed a secondary analysis of a multicenter phase 2/3, randomized, open-label trial in Europe, North and South America from 2002 to 2009. Children aged 31 days to <18 years, who were living with HIV-1 and treatment-naive, were randomized to antiretroviral therapy with two nucleoside reverse transcriptase inhibitors plus a protease inhibitor or non-nucleoside reverse transcriptase inhibitor. Time to first documented treatment disruption to any component of antiretroviral therapy, derived from treatment records and adherence questionnaires, was analyzed using Kaplan-Meier estimators and Cox proportional hazards models.

rather than the authors. Per IMPAACT policy, the data cannot be made publicly available due the ethical restrictions in the study's informed consent documents and in the International Maternal Pediatric Adolescent AIDS Clinical Trials (IMPAACT) Network's approved human subjects protection plan; public availability may compromise participant confidentiality. However, data are available to all interested researchers upon request to the IMPAACT Statistical and Data Management Center's data access committee (email address: sdac.data@fstrf.org) with the agreement of the IMPAACT Network.

**Funding:** DEY's work was supported by the Eunice Kennedy Shriver National Institute of Child Health and Human Development (NICHD) training grants to the Department of Pediatrics, Duke University Medical Center (T32 HD043029) and the Division of Infectious Diseases, Department of Pediatrics, Duke University Medical Center (T32 HD060558). The PENPACT-1 trial was sponsored jointly by the Paediatric European Network for Treatment of AIDS (PENTA) Foundation, Agènce Nationale de Recherche sur le Sida et les hepatitis virales (ANRS) and the Pediatric AIDS Clinical Trials Group (PACTG), subsequently named the International Maternal Pediatric Adolescent AIDS Clinical Trials Group (IMPAACT). Overall support for PACTG/IMPAACT was provided by the National Institute of Allergy and Infectious Diseases (NIAID) [U01 AI068632], the Eunice Kennedy Shriver National Institute of Child Health and Human Development (NICHD), and the National Institute of Mental Health (NIMH) [AI068632]. The content is solely the responsibility of the authors and does not necessarily represent the official views of the NIH. This work was supported by the Statistical and Data Analysis Center at Harvard School of Public Health, under the National Institute of Allergy and Infectious Diseases cooperative agreement #5 U01 AI41110 with the PACTG and #1 U01 AI068616 with the IMPAACT Group. Support of the sites was provided by the National Institute of Allergy and Infectious Diseases (NIAID) and the NICHD International and Domestic Pediatric and Maternal HIV Clinical Trials Network funded by NICHD (contract number N01-DK-9-001/HHSN267200800001C). PENTA is a coordinated action of the European Commission/European Union, supported by the seventh framework programme (FP7/2007-2013) under the Eurocoord grant agreement number 260694, the sixth framework contract number LSHP-CT-2006-018865, the fifth framework programme contract number QLK2-CT-2000-00150, and by the PENTA Foundation. UK clinical sites were supported by a grant from the MRC; those in Italy by a grant from

## Results

The modified intention-to-treat analysis included 263 participants. Seventy-two percent ($n =$ 190) of participants experienced at least one treatment disruption during study. At 4 years, treatment disruption probabilities were 70% (protease inhibitor) vs. 63% (non-nucleoside reverse transcriptase inhibitor). The unadjusted hazard ratio (HR) for treatment disruptions comparing protease inhibitor vs. non-nucleoside reverse transcriptase inhibitor-based regimens was 1.19, 95% confidence interval [CI] 0.88–1.61 (adjusted HR 1.24, 95% CI 0.91–1.68). By study end, treatment disruption probabilities converged (protease inhibitor 81%, non-nucleoside reverse transcriptase inhibitor 84%) with unadjusted HR 1.11, 95% CI 0.84–1.48 (adjusted HR 1.13, 95% CI 0.84–1.50). Reported reasons for treatment disruptions suggested that participants on protease inhibitors experienced greater tolerability problems.

## Conclusions

Children had similar time to treatment disruption for initial protease inhibitor and non-nucleoside reverse transcriptase inhibitor-based antiretroviral therapy, despite greater reported tolerability problems with protease inhibitor regimens. Initial pediatric antiretroviral therapy with either a protease inhibitor or non-nucleoside reverse transcriptase inhibitor may be acceptable for maintaining optimal, continuous therapy.

## Introduction

Globally, 1.8 million children are living with HIV, and 110,000 die annually due to AIDS-related illnesses [1]. For HIV-infected children, greatest survival outcomes can be achieved only with optimal, uninterrupted treatment on effective antiretroviral therapy (ART). Treatment disruptions, defined as any interruption or alteration of initial ART, may result from patient-level factors (*e.g.*, poor adherence, drug intolerance), provider-level factors (*e.g.*, prescription stops, changes, or errors), or systems-level factors (*e.g.*, stock outs, interruptions in drug delivery). Unfortunately, treatment disruptions may result in treatment failure, acquisition of resistance mutations, and loss of future treatment options—which are particularly consequential in children. Compared with adults, children have greater pharmacokinetic variability and fewer available licensed drugs [2, 3]. Due to longer lifetime antiretroviral exposure, children have more potential for long-term toxicity [4, 5]. Children have greater social vulnerability related to their dependence on others for medical care and medication administration [6, 7]. If inadequately treated, children progress much faster to AIDS and death [8–10]. As children's initial ART regimens are often their best opportunity for effective, tolerable treatment, optimizing the time on a successful initial regimen may result in greater long-term effectiveness of ART and more lifetime treatment options [11]. Analyzing longitudinal relationships between pediatric ART regimens and time to treatment disruption allows identification of initial ART regimens that pose greater challenges to maintaining optimal, continuous ART.

When deciding which regimen to prescribe to optimize clinical outcomes, clinicians must consider both drug pharmacology and potential adherence to ART regimens [12]. Boosted protease-inhibitor (PI)-based regimens appear more forgiving of treatment disruptions than do non-nucleoside reverse transcriptase inhibitor (NNRTI)-based regimens [13–17]. However, certain PI characteristics decrease adherence and tolerability, particularly in children:

the Istituto Superiore di Sanita—Progetto Terapia Antivirale 2004, 2005. GSK and BMS provided drugs in Romania. The trial was coordinated by four trials centers: the Medical Research Council (MRC) Clinical Trials Unit, London, UK (with support from the MRC); INSERM SC10, Paris, France (supported by ANRS); Frontier Science, New York, USA; and Westat, Maryland, USA (supported by NICHD). CEG's salary was partially supported by NICHD K24HD069204. Although the parent trial was overseen by members of the NIH and PENTA on the protocol team, the funders had no role in the study design, data analysis, decision to publish, or preparation of the manuscript for the secondary analysis.

**Competing interests:** Although the authors have no financial competing interests for this manuscript, DEY has been an investigator on studies unrelated to HIV supported by Astellas, Chimerix, Merck, Pfizer, Viracor-Eurofins, Kansas City Area Life Sciences Institute, and the Marion Merrell Dow Fund; DEY was a founder and is an unpaid advisory board member for the non-profit Maipelo Children's Trust/Cover the Globe, which provides HIV services in Botswana. REM has volunteered as an unpaid member of several Gilead data safety and monitoring boards, which are unrelated to this study. The other authors report no conflicts of interests. This does not alter our adherence to all PLOS ONE policies on sharing data and materials.

poor taste; gastrointestinal toxicity; and regimen complexity, such as pill burden, storage requirements, and dosing frequency [7, 17–22]. Prior pediatric studies that have assessed the ability of children to maintain continuous therapy did not do so in settings in which use of PI- vs. NNRTI-based ART regimens was randomly allocated, nor have prior studies measured treatment disruptions longitudinally. As a result, these previously conducted studies have potential for residual confounding from unmeasured covariates. Furthermore, most studies have isolated analyses of prescription patterns, adherence, and tolerability, rather than evaluating the total effect of the regimen on maintaining optimal, continuous therapy. In the PEN-PACT-1 study, 266 HIV-1-infected, treatment-naïve children from Europe, North America, and South America were randomized to ART with either a PI or NNRTI and followed longitudinally for at least 4 years [23]. We aimed to assess PENPACT-1 participants for differences in time to treatment disruption across randomized PI vs. NNRTI treatment arms at 4 years and end of study.

## Methods

### Study design and participants

PENPACT-1 (Paediatric European Network for Treatment of AIDS [PENTA] 9 / Pediatric AIDS Clinical Trials Group [PACTG] 390) was an international multicenter phase 2/3, randomized, open-label trial enrolling children living with HIV-1 from 68 clinical centers in 13 countries in Europe and North and South America between September 25, 2002, and September 7, 2005 (S1 Protocol) [23]. Eligible children aged 31 days to less than 18 years were HIV-1-infected and had not received ART or received only antiretrovirals for <56 days to reduce mother-to-child transmission (excluding single-dose nevirapine). All parents or guardians and children, as appropriate, gave written consent for the parent trial; this protocol was conducted in accordance with the Declaration of Helsinki and approved by the relevant ethics committee or institutional review board (IRB) for each participating center. The secondary analysis on time to treatment disruption was performed under a data request and was reviewed only at IRBs where the analysis was performed. The secondary analysis was deemed exempt by the Duke University IRB and approved by the University of North Carolina-Chapel Hill and Children's Mercy Kansas City IRBs. This study is registered with the International Standard Randomised Controlled Trial Number Registry (ISRCTN73318385) at https://doi.org/10.1186/ISRCTN73318385 and ClinicalTrials.gov (NCT00039741) at https://clinicaltrials.gov/ct2/show/NCT00039741.

Children were randomized 1:1 to start ART with two nucleoside reverse transcriptase inhibitors (NRTIs) plus either a PI or NNRTI. Randomization was stratified by age (<3 years or ≥3 years); receipt of perinatal ART prophylaxis; and research network (PENTA or PACTG), which varied by region; with variable block sizes. The study was open label, and the treating clinician chose the two NRTI drugs combined with a drug from the randomly assigned PI or NNRTI class. Children underwent clinical and HIV-1 RNA viral load assessments at randomization (week 0), weeks 2, 4, 8, 12, 16, 24, and then every 12 weeks until the last child assigned to treatment reached 4 years of follow-up (August 31, 2009). Treatment starts, changes, and stoppages were recorded at these clinical visits and *ad hoc* throughout the study. Trained study personnel administered validated adherence questionnaires every 24 weeks after randomization, or if missed, at the following attended visit [24]. Adherence questionnaires were harmonized across networks to collect key data. Specifically, adherence questionnaires recorded the number of missed doses to all antiretrovirals over the 3 days prior to these 24-weekly visits and barriers to adherence experienced within 2 weeks prior to these

visits. Four years of follow-up was defined as the week 192 visit plus a 6-week lag to capture late visits.

## Outcomes

We defined time to treatment disruption as the number of weeks between randomization and the first documented treatment disruption event. We defined treatment disruption as stopping, switching, or reporting missed doses of any component of the initial ART regimen for any reason except recall of nelfinavir (June 2007) or planned treatment interruptions. Stopping was defined as any duration of treatment discontinuation, regardless of whether treatment was restarted or changed in the future, whereas switches were defined as immediate changes of therapy. Information on ART stoppages or switches was derived from participants' treatment records, and missed doses were defined as any questionnaire-reported missed doses within 3 days prior to the study visit.

Additional analyses included adjustment for stratified randomization factors (age, receipt of perinatal ART prophylaxis, research network), assessed differences in outcome for the primary follow-up time point (4 years) vs. the entire study, and explored reasons for treatment disruptions. Reasons for treatment disruptions were analyzed using (1) the treatment record's documented rationale for ART stop or change and (2) any questionnaire-reported barriers to adherence within 2 weeks prior to the visit when missed dose(s) were reported. Only one reason for treatment disruption was allowed on the treatment record; thus a single response, such as "caregiver request," may not exclude additional reasons. Multiple reasons were allowed on adherence questionnaires.

We assessed the sensitivity of our results to our definition of treatment disruption. Our alternative outcome definitions included restricting treatment record-based treatment disruptions (or any questionnaire event) to drug changes or stops lasting more than 3 days or 14 days and restricting treatment record-based treatment disruptions (or any questionnaire event) to only events including the PI or NNRTI drug component.

## Statistical analysis

PI vs. NNRTI treatment groups were assessed according to a modified intention-to-treat (mITT) analysis consistent with the original study [23]. The sole modification was removal of three participants: two who withdrew consent prior to ART initiation, and one with a major eligibility violation. Follow-up began at date of randomization. Participants were right-censored for initial treatment contrary to randomization, planned treatment interruption, death, withdrawal of consent, loss to follow-up, or study end.

For the primary outcome, we estimated the risk of treatment disruptions using the complement of the Kaplan-Meier estimator. We estimated the hazard ratio for treatment disruptions using Cox proportional hazards models. Proportional hazards assumptions were assessed graphically, using time-interaction terms, and with martingale residuals. In adjusted analyses, we stratified by baseline randomized stratification variables: age, exposure to perinatal ART, and research network. Analyses were conducted in SAS® version 9.4 (Cary, NC).

## Results

PENPACT-1 enrolled 266 HIV-1 infected children from 68 centers in 13 countries in Europe, North America, and South America. The mITT analysis was restricted to 263 participants who initiated ART. Participants were a median age of 6.5 years at enrollment (IQR [interquartile range], 1.8–12.9), 52% male, 49% black, and 79% exposed to HIV via vertical transmission (Table 1). Fifty-one percent had moderate to severe clinical symptoms (CDC stage B or C).

Median growth parameters were below average (weight-for-age Z score -0.6; height-for-age Z score -0.9). Median CD4 Z-score was -3.5, consistent with predominance of moderate to severe immunosuppression, and median viral load was 5.0 $\log_{10}$ copies/mL. Whereas 15% of children had ART exposure for prevention of mother-to-child transmission, 4% had at least one major resistance mutation at baseline. Although treatment groups had differences in racial distribution, baseline characteristics relating to mode of HIV-1 acquisition, clinical and immunological status, and ART resistance were generally balanced across ART regimens, consistent with the randomized design.

Median follow-up time was 261 weeks (IQR, 217–313). Two participants in each arm were started on a PI or NNRTI contrary to randomization; two underwent planned treatment interruption; five withdrew from study after ART initiation; 37 were lost to follow-up; and one patient died, due to HIV-related complications (Fig 1). Two hundred forty-nine participants ever completed an adherence questionnaire, totaling 2,112 questionnaires over the duration of the study for a mean of 8.5 questionnaires per participant.

Overall, 191 of 263 participants had at least one treatment disruption event during the study, with 66% (95% confidence interval [CI] 61–72%) treatment disruption probability at 4 years (primary follow-up period) and 83% (95% CI 76–91%) treatment disruption probability at study end (6.5 years). At 4 years, probabilities of treatment disruption were 70% (95% CI 62–78%) vs. 63% (95% CI 55–72%) in the PI and NNRTI arms, respectively (Fig 2). Hazards for treatment disruption, however, were similar for PI vs. NNRTI-based regimens (unadjusted hazard ratio [HR] 1.19, 95% CI 0.88–1.61), even after adjustment for stratification factors of age, receipt of perinatal ART, and research network/region (adjusted HR 1.24, 95% CI 0.91–1.68).

After 4 years, treatment disruption probabilities converged, such that treatment disruption probabilities at study end were 81% (95% CI 72–90%) for PI vs. 84% (95% CI 73–94%) for NNRTI arms, but changes over time in the hazard ratio of treatment disruption by treatment arms were non-significant (unadjusted *P* for interaction = 0.33, adjusted *P* = 0.21). Hazards for treatment disruption over the entire study period were similar for PI vs. NNRTI-based regimens, unadjusted (HR 1.11, 95% CI 0.84–1.48) and adjusted (HR 1.13, 95% CI 0.84–1.50).

Of 191 treatment disruption events, 126 events were based on ART regimen stoppages or changes in the treatment record, and 67 events were reported missing doses on adherence questionnaires, with two participants experiencing both event types simultaneously. Of the treatment stops or changes, 25% of events were substitutions of at least one first-line ART drug (PI 32%, NNRTI 16%), 53% were stoppage or suspension of the entire first-line ART regimen (PI 48%, NNRTI 59%), and 22% were switches to a second-line ART regimen (PI 20%, NNRTI 25%). Most frequent reasons documented for ART stops or changes were adverse events (34%), viral failure (22%), caregiver request (18%), non-adherence (7%), and temporary break (6%), with the greatest difference between PIs over NNRTIs for adverse events (Table 2).

Reports of missed doses on adherence questionnaires were balanced between PI and NNRTI arms, as 35% of non-adherence events in each arm were from patient or caregiver reports. The most common questionnaire-reported barriers to adherence, forgetting/lacking support (30%) or running out of medications (25%), were balanced between PI and NNRTI regimens. Other common questionnaire-reported adherence problems—including difficulties with administration, such as those attributed to intolerance, taste, patient refusal (24%); fear of disclosure to others (22%); patient refusal (21%); difficulties with scheduling or lifestyle (18%); and concerns about drug toxicity (16%)—were more frequently reported in participants in the PI arm (Table 2).

In sensitivity analyses, modifications of the outcome definition did not result in substantial hazard ratio changes. Point estimates at 4 years remained similar to the primary analysis when

**Table 1. Baseline characteristics of study participants according to initial ART regimen.**

| Variable | | PI | NNRTI | Total |
|---|---|---|---|---|
| | | | Randomized Group | |
| N | | 131 | 132 | 263 |
| Age | | | | |
| <3 years | n (%) | 34 (26%) | 36 (27%) | 70 (27%) |
| 3–17 years | n (%) | 97 (74%) | 96 (73%) | 193 (73%) |
| Age in years | Median (IQR) | 7.1 (2.8, 13.7) | 6.4 (2.7, 11.0) | 6.5 (2.8, 12.9) |
| Sex | | | | |
| Male | n (%) | 69 (53%) | 67 (51%) | 136 (52%) |
| Race | | | | |
| Black, Non-Hispanic | n (%) | 60 (46%) | 69 (52%) | 129 (49%) |
| White, Non-Hispanic | n (%) | 40 (31%) | 29 (22%) | 69 (26%) |
| Hispanic/Other | n (%) | 31 (24%) | 34 (26%) | 65 (25%) |
| Research Network[a] | | | | |
| PENTA | n (%) | 95 (73%) | 93 (70%) | 188 (71%) |
| PACTG/IMPAACT | n (%) | 36 (27%) | 39 (30%) | 75 (29%) |
| Route of Infection | | | | |
| Vertical | n (%) | 103 (79%) | 106 (80%) | 209 (79%) |
| Other/Unknown | n (%) | 28 (21%) | 26 (20%) | 54 (21%) |
| CDC Clinical Stage | | | | |
| N | n (%) | 27 (21%) | 29 (22%) | 56 (21%) |
| A | n (%) | 35 (27%) | 37 (28%) | 72 (27%) |
| B | n (%) | 41 (31%) | 43 (33%) | 84 (32%) |
| C | n (%) | 28 (21%) | 23 (17%) | 51 (19%) |
| Weight-for-Age Z-score | Median (IQR) | -0.5 (-1.6, 0.1) | -0.7 (-1.6, 0.2) | -0.6 (-1.6, 0.1) |
| Height-for-Age Z-score | Median (IQR) | -0.9 (-1.5, -0.2) | -0.9 (-1.8, 0) | -0.9 (-1.7, -0.2) |
| CD4 Z score | Median (IQR) | -3.6 (-7.2, -1.7) | -3.4 (-6.5, -1.4) | -3.5 (-6.8, -1.6) |
| Viral Load $\log_{10}$ copies/mL | Median (IQR) | 5.1 (4.5, 5.7) | 5.0 (4.5, 5.6) | 5.0 (4.5, 5.7) |
| Perinatal ART Exposure | n (%) | 19 (15%) | 20 (15%) | 39 (15%) |
| ≥1 Major Resistance Mutation[b] | n/N (%) | 5/116 (4%) | 5/123 (4%) | 10/239 (4%) |
| HIV-1 subtype | | | | |
| B | n (%) | 52 (42%) | 49 (39%) | 101 (41%) |
| C | n (%) | 13 (11%) | 12 (10%) | 25 (10%) |
| F | n (%) | 25 (20%) | 23 (18%) | 48 (19%) |
| A/CRF_AG/D/G | n (%) | 21 (17%) | 31 (25%) | 52 (21%) |
| Unclassified | n (%) | 12 (10%) | 11 (9%) | 23 (9%) |
| Switching Threshold | | | | |
| 1,000 copies/mL | n (%) | 66 (50%) | 68 (52%) | 134 (51%) |
| 30,000 copies/mL | n (%) | 65 (50%) | 64 (48%) | 129 (49%) |
| Duration of Follow-Up in weeks | Median (IQR) | 263 (217, 313) | 260 (219, 316) | 261 (217, 313) |

ART, antiretroviral therapy; IQR, interquartile range; N, total sample size; n, subsample size; NNRTI, non-nucleoside reverse transcriptase inhibitor; PACTG, Pediatric AIDS Clinical Trials Group; PENTA, Paediatric European Network for Treatment of AIDS; PI, protease inhibitor.

[a] PENTA sites were predominantly in Europe, South America, and the Bahamas. PACTG sites were based primarily in the United States.

[b] Not all patients had successful baseline genotypic resistance assays.

restricting events on the treatment record (or any event on questionnaire) to only ART regimen stops or changes lasting >3 days (unadjusted HR 1.16, 95% CI 0.85–1.57; adjusted HR 1.19, 95% CI 0.88–1.63), only ART regimen stops or changes lasting >14 days (unadjusted HR

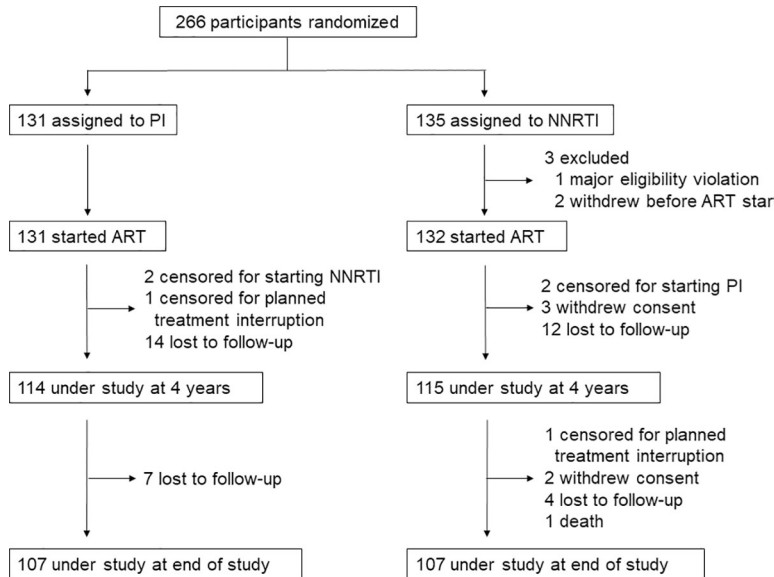

**Fig 1. Study profile.** ART, antiretroviral therapy; NNRTI, non-nucleoside reverse transcriptase inhibitor; PI, protease inhibitor.

1.27, 95% CI 0.93–1.74; adjusted HR 1.32, 95% CI 0.96–1.81), or only stops or changes including the PI or NNRTI drug (unadjusted HR 1.14, 95% CI 0.84–1.55; adjusted HR 1.18, 95% CI 0.87–1.61).

## Discussion

In PENPACT-1, our estimates were not compatible with large differences in time to treatment disruption between participants randomized to PIs versus NNRTIs. Point estimates were mildly in the direction of more treatment disruptions in PI-based regimens, particularly in the primary end point of 4 years, but differences were small, possibly due to chance, and appeared to decrease by study end. Exploration of reasons for treatment disruptions suggested that PI-based regimens may be less tolerable, both due to adverse events leading to treatment stoppages or substitutions and to regimen-specific adherence barriers reported on the adherence questionnaire. However, these PI-associated difficulties did not interrupt continuous therapy to the initial PI-based regimens more than they did to NNRTI-based regimens.

Although we did not find a meaningful difference in treatment disruptions in PI vs. NNRTI-based regimens, the secondary analyses exploring reasons for treatment disruptions suggested that administration of a PI-based regimen to a child may be a struggle, even if not resulting in actual missed doses. The treatment record suggested that participants experienced more adverse events to PIs over NNRTIs, but adherence questionnaire responses formed a pattern of difficulties with PI tolerability, whether attributed to taste, medication volume or pill burden, toxicity, or simply patient refusal. This pattern would be consistent with existing literature on PI vs. NNRTI regimens. PIs have higher drug toxicity, especially gastrointestinal side effects, and intolerance, particularly regarding their noxious taste [7, 18–20, 25–27]. Even if children are able to swallow pills, certain PIs are available only as large pills [28, 29]. At the time of PENPACT-1, no PIs were available as complete-regimen combinations for children, whereas single-tablet NNRTI regimens could facilitate adherence through administration of fewer pills [2, 30–34]. More recently, a novel four-in-one fixed-dose combination of abacavir, lamivudine, and LPV/r granule-filled capsules has been under study and submitted to the

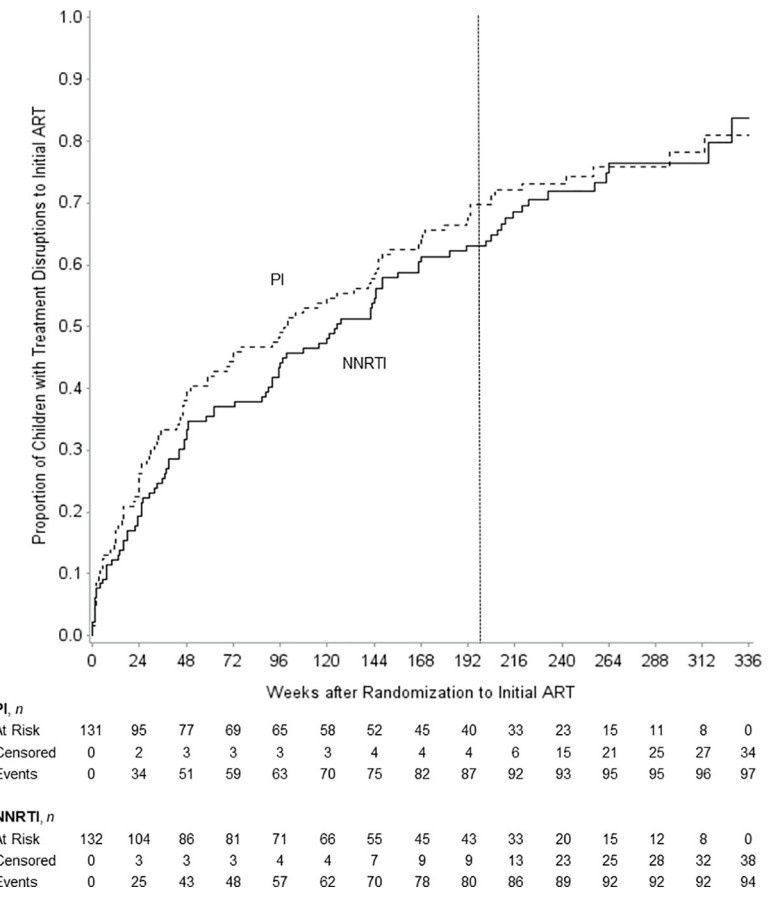

**Fig 2. Proportion of children experiencing treatment disruption from initial ART regimen by study week.** The vertical line delineates 4 years on study. ART, antiretroviral therapy; *n*, subsample size; NNRTI, non-nucleoside reverse transcriptase inhibitor; PI, protease inhibitor.

FDA for approval [35]. Participants reported more barriers to adherence in PIs related to scheduling or lifestyle interference, which may relate to dosing frequency. We hypothesize that increased fear of disclosure to others, as noted in the PI arm, may relate to difficulties concealing drug administration when given more frequently. Higher dosing frequency has been associated with more frequent treatment disruptions [20, 30, 33, 34, 36–39]. Some NNRTIs, most notably efavirenz, have more suitable pharmacokinetics for once daily administration. In our study, most PI-based regimens were administered at least twice daily, whereas some commonly used NNRTI-based regimens allowed once-daily dosing.

Most children in PENPACT-1 experienced a treatment disruption event during the study. Only about one-third of participants remained continuously on their initial ART at 4 years; only one-sixth remained continuously on initial ART at study end. These results are consistent with other pediatric data on the durability of first-line treatment regimens [40]. Maintaining continuous therapy on ART is critical to sustained HIV-related outcomes, as suppressing viral load decreases the probability of HIV sub-populations acquiring antiretroviral resistance mutations and chances of forward infection [41–49]. Although optimal adherence targets vary by PI vs. NNRTI class, adherence has been modest across ART studies, especially patients failing to achieve viral suppression [12, 16, 17, 30, 50–56]. Notably, ART appears to be less successful in producing viral suppression in children, who are more prone to viral failure and

**Table 2. Reasons listed for treatment disruption events.**

| Reason / Barrier | | PI | NNRTI | Total |
|---|---|---|---|---|
| Treatment Record[a] | | | | |
| Adverse event | n (%) | 24 (37%) | 19 (31%) | 43 (34%) |
| Viral failure | n (%) | 13 (20%) | 15 (25%) | 28 (22%) |
| Caregiver request | n (%) | 11 (17%) | 12 (20%) | 23 (18%) |
| Non-adherence | n (%) | 6 (9%) | 3 (5%) | 9 (7%) |
| Temporary break | n (%) | 3 (5%) | 5 (8%) | 8 (6%) |
| Unknown | n (%) | 5 (8%) | 1 (2%) | 6 (5%) |
| Drug supply problem | n (%) | 1 (2%) | 2 (3%) | 3 (2%) |
| Intercurrent illness | n (%) | 0 (0%) | 2 (3%) | 2 (2%) |
| Resistance | n (%) | 1 (2%) | 1 (2%) | 2 (2%) |
| Parent forgot | n (%) | 1 (2%) | 0 (0%) | 1 (1%) |
| Simplification | n (%) | 0 (0%) | 1 (2%) | 1 (1%) |
| Treatment record total | n | 65 | 61 | 126 |
| Adherence Questionnaire[b] | | | | |
| Forgot/lack of support | n (%) | 10 (29%) | 10 (30%) | 20 (30%) |
| Ran out of drug | n (%) | 8 (24%) | 9 (27%) | 17 (25%) |
| Problems taking some of the drugs (*e.g.*, intolerance, taste, medication volume) | n (%) | 11 (32%) | 5 (15%) | 16 (24%) |
| Fear of disclosure to others | n (%) | 10 (29%) | 5 (15%) | 15 (22%) |
| Patient refused/didn't want to take drugs | n (%) | 10 (29%) | 4 (12%) | 14 (21%) |
| Scheduling/lifestyle interference | n (%) | 9 (26%) | 3 (9%) | 12 (18%) |
| Drug toxicity concerns | n (%) | 7 (21%) | 4 (12%) | 11 (16%) |
| Supervised by someone else or multiple caregivers | n (%) | 6 (18%) | 5 (15%) | 11 (16%) |
| Patient unwell | n (%) | 6 (18%) | 4 (12%) | 10 (15%) |
| Other | n (%) | 4 (12%) | 5 (15%) | 9 (13%) |
| Different routine/change in living situation | n (%) | 3 (9%) | 4 (12%) | 7 (10%) |
| Fed up giving/taking drugs | n (%) | 3 (9%) | 2 (6%) | 5 (7%) |
| Think medication is not needed or not helping | n (%) | 2 (6%) | 2 (6%) | 4 (6%) |
| Caregiver unwell/depressed | n (%) | 0 (0%) | 0 (0%) | 0 (0%) |
| Total listed problems on questionnaire[b] | n | 89 | 62 | 151 |
| Total participants with questionnaire-reported missed doses | n | 34 | 33 | 67 |
| Total Treatment Disruption Events[c] | n | 97 | 94 | 191 |

*n*, subsample size or number of events; NNRTI = non-nucleoside reverse transcriptase inhibitor; PI = protease inhibitor.

[a] One category allowed per treatment record change or stop.

[b] Participants may have answered in more than one category.

[c] Some participants had both a treatment record and adherence questionnaire event at the same time.

resistance due to higher plasma viral loads, less robust antiviral immune responses, greater pharmacokinetic variability, and social dependency [44, 57]. Adolescents have particularly worse viral and immunological outcomes, due to poor ART adherence [48, 49, 58–60]. The large proportion of children in PENPACT-1 with disruptions of their initial ART raises concerns regarding long-term durability, especially as these patients were receiving adherence support on a clinical trial protocol at specialty pediatric HIV centers.

Based on our data, choice of an initial PI- vs. NNRTI-based regimen may not have a major impact on ART treatment disruption. Despite differences in reported regimen-related adherence barriers, participants in both treatment arms persevered in taking their regimens similarly. Moreover, the most common questionnaire-reported barriers were not regimen-specific:

forgetting/lack of support and running out of drug. Novel interventions may still be able to improve the experience of drug administration. Pediatric pellets are heat-stable and generally more acceptable than syrups, but palatability and administration problems persist and may increase over time [61–63]. Pediatric granules, especially in the four-in-one combination, may improve palatability and decrease pill burden [35, 64]. Precision medicine related to taste-sensing genotypes may hold promise for prescribing according to individualized palatability [65]. In adult data, integrase strand transferase inhibitors (INSTIs) have been at least as tolerable as PIs or NNRTIs, if not more so, and INSTIs are increasingly preferred drugs in children [66–69]. Nevertheless, a primary goal of optimizing continuous therapy to ART is durable viral suppression, which was comparable across PI vs. NNRTI arms in this study's parent trial, although similar trials had variable results [23, 70–74]. In this study population, choice of either PI- or NNRTI-based initial ART appears acceptable.

Our estimates of treatment disruption may have had measurement error. First, we had no direct measures of drug exposure, such as therapeutic drug monitoring. Treatment records captured only prescribing events and documented ART disruptions, and the adherence questionnaires relied on accurate reporting by either the child or the caregiver, if present and willing to answer. Although we relied on a questionnaire that has previously been validated [24], reporting biases and unanswered questionnaires may have affected our measures of missed doses. Our combining treatment records and adherence questionnaires into a composite outcome should have decreased measurement error from either instrument individually. Second, adherence questionnaires in this study focused on ART adherence over the 3 days prior to the most recent visit and inquired about adherence barriers encountered over the prior 2 weeks, rather than a daily measure of adherence throughout the study. The time-varying nature of treatment disruption means that patients may have experienced an initial or temporary period of treatment disruption that was subsequently corrected [75, 76], but our analysis presents only data on time to first event of treatment disruption. Third, limited participant report of individual drugs missed on the adherence questionnaire precluded definitive identification of treatment disruptions of individual drugs. Instead, we assessed treatment disruption to any component of the ART regimen. Fourth, heterogeneity of adherence questionnaires across networks, ages, and respondents regarding barriers to therapy should caution against rigorous interpretation of reasons for treatment disruptions. Finally, this study size was not sufficient to distinguish differences on the order of 7%, as was seen at 4 years.

## Conclusions

In conclusion, children in PENPACT-1 had similar time to treatment disruption for initial PI-based regimens and NNRTI-based regimens. Although secondary analyses suggest that PI-based regimens may be more difficult to tolerate and may be less convenient to administer, these difficulties did not result in a large difference in children stopping, changing, or missing doses at 4 years (PI 70%, NNRTI 63%), and any suggested differences diminished by study end (PI 81%, NNRTI 84%). Initial ART with either a PI or NNRTI may be acceptable for maintaining continuous therapy on ART in children.

## Supporting information

**S1 Checklist. CONSORT checklist.**
(DOC)

**S1 Protocol. PENPACT-1 trial protocol.**
(PDF)

## Acknowledgments

**The PENPACT-1 (PENTA 9 / PACTG 390) study team**

**PENPACT-1 protocol team.** PACTG/IMPAACT/NICHD: P Brouwers, D Costello, E Ferguson, S Fiscus, J Hodge, M Hughes, C Jennings, A Melvin (Co-Chair), R McKinney (Co-Chair), L Mofenson, M Warshaw, ME Smith, S Spector, E Stiehm, M Toye, R Yogev.

PENTA: JP Aboulker, A Babiker, H Castro, A Compagnucci, A De Rossi, C Giaquinto, J Darbyshire, M Debré, DM Gibb, L Harper, L Harrison, N Klein, D Pillay, Y Saidi, G Tudor-Williams (Co-Chair), AS Walker.

Data and Safety Monitoring Board: B Brody, C Hill, P Lepage, J Modlin, A Poziak, M Rein (Chair 2002–2003), M Robb (Chair 2004–2009), T Fleming, S Vella, KM Kim.

**Clinical sites (L = lab, P = pharmacy).** Argentina: Hospital de Pediatria Dr JP Garrahan, Buenos Aires: R Bologna, D Mecikovsky, N Pineda, L Sen (L), A Mangano (L), S Marino (L), C Galvez (L); Laboratorio Fundai: G Deluchi (L).

Austria: Universitätsklinik für Kinder und Jugendheilkunde, Graz: B Zöhrer, W Zenz, E Daghofer, K Pfurtscheller, B Pabst (L).

Bahamas: Princess Margaret Hospital: MP Gomez, P McNeil, M Jervis, I Whyms, D Kwolfe, S Scott (P).

Brazil: University of Sao Paulo at Ribeirao Preto: MM Mussi-Pinhata, ML Issac, MC Cervi, BVM Negrini, TC Matsubara, C BSS de Souza (L), JC Gabaldi (P); Institute of Pediatrics (IPPMG), Federal University of Rio de Janeiro: RH Oliveira, MC Sapia, T Abreu, L Evangelista, A Pala, I Fernandes, I Farias, M de F Melo (L), H Carreira (P), LM Lira (P); Instituto de Infectologia Emilio Ribas, Sao Paolo: M della Negra, W Queiroz, YC Lian; DP Pacola; Fleury Laboratories; Federal University of Minas Gerais, Belo Horizonte: J Pinto, F Ferreira, F Kakehasi, L Martins, A Diniz, V Lobato, M Diniz, C Hill (L), S Cleto (L), S Costa (P), J Romeiro (P).

France: Hôpital d'enfants Armand Trousseau, Paris: C Dollfus, MD Tabone, MF Courcoux, G Vaudre A Dehée (L), A Schnuriger (L), N Le Gueyades (P), C De Bortoli (P); CHU Hôtel Dieu, Nantes: F Méchinaud, V Reliquet, J Arias (L), A Rodallec (L), E André (L), I Falconi (P), A Le Pelletier (P); Hôpital de l'Archet II, Nice, F Monpoux, J Cottalorda (L), S Mellul (L); Hôpital Jean Verdier, Bondy: E Lachassinne; Laboratoire de virologie-Hôpital Necker Enfants Malades, Paris: J Galimand (L), C Rouzioux (L), ML Chaix (L), Z Benabadji (P), M Pourrat (P); Hôpital Cochin Port-Royal- Saint Vincent de Paul, Paris: G Firtion, D Rivaux, M Denon, N Boudjoudi, F Nganzali, A Krivine (L), JF Méritet (L), G Delommois (L), C Norgeux (L), C Guérin (P); Hôpital Louis Mourier, Colombes: C Floch, L Marty, H Hichou (L), V Tournier (P); Hôpital Robert Debré, Paris: A Faye, I Le Moal, M Sellier (P), L Dehache (P); Laboratoire de virologie-Hôpital Bichat Claude Bernard-Paris: F Damond (L), J Leleu (L), D Beniken (L), G Alexandre-Castor (L).

Germany: Universitäts-Kinderklinik Düsseldorf: J Neubert, T Niehues, HJ Laws, K Huck, S Gudowius, K Siepermann, H Loeffler, S Bellert (L), A Ortwin (L); Universitäts-Kinderkliniken, Munich: G Notheis, U Wintergerst, F Hoffman, A Werthmann, S Seyboldt, L Schneider, B Bucholz; Charité-Medizische Fakultät der Humboldt-Universität zu Berlin: C Feiterna-Sperling, C Peiser, R Nickel, T Schmitz, T Piening, C Müller (L); Kinder und Jugendklinik, Universität Rostock: G Warncke, M Wigger, R Neubauer.

Ireland: Our Lady's Children's Hospital Crumlin, Dublin: K Butler, AL Chong, T Boulger, A Menon, M O'Connell, L Barrett, A Rochford, M Goode, E Hayes, S McDonagh, A Walsh, A Doyle, J Fanning (P), M O'Connor (P), M Byrne (L), N O'Sullivan (L), E Hyland (L).

Italy: Clinica Pediatrica, Ospedale L Sacco, Milan: V Giacomet, A Viganò, I Colombo, D Trabattoni (L), A Berzi (L); Clinica Pediatrica, Università di Brescia: R Badolato, F Schumacher, V Bennato, M Brusati, A Sorlini, E Spinelli, M Filisetti, C Bertulli; Clinica Pediatrica,

Università di Padova: O Rampon, C Giaquinto, M Zanchetta (L); Ospedale S. Chiara, Trento: A Mazza, G Stringari, G Rossetti (L); Ospedale del Bambino Gesù, Rome: S Bernardi, A Martino, G Castelli Gattinara, P Palma, G Pontrelli, H Tchidjou, A Furcas, C Frillici, A Mazzei, A Zoccano (P), C Concato (L).

Romania: Spitalul Clinic de Boli Infectioase Victor Babes, Bucharest: D Duiculescu, C Oprea, G Tardei (L), F Abaab (P); Institutul de Boli Infectioase Matei Bals, Bucharest: M Mardarescu, R Draghicenoiu, D Otelea (L), L Alecsandru (P); Clinic Municipal, Constanta: R Matusa, S Rugina, M Ilie, Silvia Netescu (P). Clinical monitors: C Florea, E Voicu, D Poalelungi, C Belmega, L Vladau, A Chiriac.

Spain: Hospital Materno-Infantil 12 de Octubre, Madrid: JT Ramos Amador, MI Gonzalez Tomé, P Rojo Conejo, M Fernandez, R Delgado Garcia (L), JM Ferrari (P); Institute de Salud Carlos III, Madrid: M Garcia Lopez, MJ Mellado Peña, P Martin Fontelos, I Jimenez Nacher (P); Biobanco Gregorio Marañon, Madrid: MA Muñoz Fernandez (L), JL Jimenez (L), A García Torre (L); clinical monitors: M Penin, R Pineiro Perez, I Garcia Mellado.

UK: Bristol Royal Children's Hospital: A Finn, M LaJeunesse, E Hutchison, J Usher (L), L Ball (P), M Dunn (P); St. George's Healthcare NHS Trust, London: M Sharland, K Doerholt, S Storey, S Donaghy, R Chakraborty, C Wells (P), K Buckberry (P), P Rice (P); University Hospital of North Staffordshire: P McMaster, P Butler, C Farmer (L), J Shenton (P), H Haley (P), J Orendi (L), University Hospital Lewisham: J Stroobant, L Navarante, P Archer, C Mazhude, D Scott, R O'Connell, J Wong (L), G Boddy (P); Sheffield Children's Hospital: F Shackley, R Lakshman, J Hobbs, G Ball (L), G Kudesia (L), J Bane (P), D Painter (P); Ealing Hospital NHS Trust: K Sloper, V Shah, A Cheng (P), A Aali (L); King's College Hospital, London: C Ball, S Hawkins, D Nayagam, A Waters, S Doshi (P); Newham University Hospital: S Liebeschuetz, B Sodiende, D Shingadia, S Wong, J Swan (P), Z Shah (P); Royal Devon and Exeter Hospital: A Collinson, C Hayes, J King (L), K O'Connor (L); Imperial College Healthcare NHS Trust, London: G Tudor-Williams, H Lyall, K Fidler, S Walters, C Foster, D Hamadache, C Newbould, C Monrose, S Campbell, S Yeung, J Cohen, N Martinez-Allier, D Melvin, J Dodge, S Welch, G Tatum, A Gordon, S Kaye (L), D Muir (L), D Patel (P); Great Ormond Street Hospital: V Novelli, D Gibb, D Shingadia, K Moshal, J Lambert, N Klein, J Flynn, L Farrelly, M Clapson, L Spencer, M Depala (P); Institute of Child Health, London: M Jacobsen (L); John Radcliffe Hospital, Oxford: S Segal, A Pollard, D Kelly, S Yeadon, B Ohene-Kena Y Peng (L), T Dong (L), Y Peng (L), K Jeffries (L), M Snelling (P), Nottingham University Hospitals: A Smyth, J Smith; Chelsea and Westminster Hospital, London: B Ward; Mortimer Market Centre, London: E Jungmann; Doncaster Royal Infirmary: C Ryan, K Swaby; Health Protection Agency, London: A Buckton (L); Health Protection Agency, Birmingham: E Smit (L).

USA: Harlem Hospital Center: EJ Abrams, S Champion, AD Fernandez, D Calo, L Garrovillo, K Swaminathan, T Alford, M Frere, Columbia University Laboratories, J Navarra (P, Town Total Health); NYU School of Medicine: W Borkowsky, S Deygoo, T Hastings, S Akleh, T Ilmet (L); Seattle Children's Hospital: A Melvin, K Mohan, G Bowen (Additional support by NIH Grant UL1 RR025014); University of South Florida: PJ Emmanuel, J Lujan-Zimmerman, C Rodriguez, S Johnson, A Marion, C Graisbery, D Casey, G Lewis; All Children's Hospital Laboratories; Oregon Health and Science University: J Guzman-Cottrill, R Croteau; San Juan City Hospital: M Acevedo-Flores, M Gonzalez, L Angeli; L Fabregas, Lab 053, P Valentin (P); SUNY-Upstate Medical University-Syracuse: L Weiner, KA Contello, W Holz, M Butler; SUNY, Health Science Center at Stonybrook: S Nachman, MA Kelly, DM Ferraro, UNC Retrovirology Lab; Howard University Hospital: S Rana, C Reed, E Yeagley, A Malheiro, J Roa; LAC and USC Medical Center: M Neely, A Kovacs, L Spencer, J Homans, Y Rodriguez Lozano, Maternal Child Virology Research Laboratory, Investigational Drug Service; South Florida Children's Diagnostic & Treatment Center: A Puga, G Talero, R Sellers; Broward

General Medical Center, University of Miami (L); University College of Florida College of Medicine-Gainesville: R Lawrence; University of Rochester Pediatrics: GA. Weinberg, B Murante, S Laverty; Miller Children's Hospital Long Beach: A Deveikis, J Batra, T Chen, D Michalik, J Deville, K Elkins, S Marks, J Jackson Alvarez, J Palm, I Fineanganofo (L), M Keuth (L), L Deveikis (L), W Tomosada (P); Tulane University New Orleans: R Van Dyke, T Alchediak, M Silio, C Borne, S Bradford, S Eloby-Childress (L), K Nguyen (P); University of Florida/Jacksonville: MH Rathore, A Alvarez; A Mirza, S Mahmoudi, M Burke; University of Puerto Rico: IL Febo, L Lugo, R Santos; Children's Hospital Los Angeles: JA Church, T Dunaway, C Rodier; St. Jude/UTHSC: P Flynn, N Patel, S DiScenza, M Donohoe; WNE Maternal Pediatric Adolescent AIDS: K Luzuriaga, D Picard; Texas Children's Hospital: M Kline, ME Paul, WT Shearer, C McMullen-Jackson; Children's Memorial Hospital, Chicago: R Yogev, E Chadwick, E Cagwin, K Kabat; New Jersey Medical School: A Dieudonne, P Palumbo, J Johnson; Robert Wood Johnson Medical School, New Brunswick: S Gaur, L Cerracchio; Columbia IMPAACT: M Foca, A Jurgrau, S Vasquez Bonilla, G Silva; Babies' Hospital, Columbia/Presbyterian Medical Center, New York: A Gershon; University of Massachusetts Medical Center, Worcester: J Sullivan; UCLA Medical Center, Los Angeles: Y Bryson; Children's Hospital, Seattle: L Frenkel; UNC-Chapel Hill Virology Lab: S Fiscus (L), J Nelson (L).

**Trials units/support.** INSERM SC10 Paris: JP Aboulker, A Compagnucci, G Hadjou, S Léonardo, Y Riault, Y Saïdi.

MRC Clinical Trials Unit, UK: A Babiker, L Buck, JH Darbyshire, L Farrelly, S Forcat, DM Gibb, H Castro, L Harper, L Harrison, J Horton, D Johnson, S Moore, C Taylor, AS Walker.

Westat/NICHD: D Collins, S Buskirk, P Kamara, C Nesel, M Johnson, A Ferreira.

Frontier Science: J Hodge, J Tutko, H Sprenger.

IMPAACT: M Hughes, M Warshaw, P Britto, C Powell.

NIAID: R DerSimonian, E Handelsman (deceased).

PENTA Steering Committee: JP Aboulker, J Ananworanich, A Babiker, E Belfrage, S Bernardi, S Blanche, AB Bohlin, R Bologna, D Burger, K Butler, G Castelli-Gattinara, H Castro, P Clayden, A Compagnucci, JH Darbyshire, M Debré, R De Groot, M Della Negra, A De Rossi, A Di Biagio, D Duiculescu, A Faye, V Giacomet, C Giaquinto (Chair), DM Gibb, I Grosch-Wörner, M Hainault, L Harper, N Klein, M Lallemant, J Levy, H Lyall, M Marczynska, M Mardarescu, MJ Mellado Pena, D Nadal, L Naver, T Niehues, C Peckham, D Pillay, J Popieska, JT Ramos Amador, L Rosado, R Rosso (deceased), C Rudin, Y Saïdi, H Scherpbier, M Sharland, M Stevanovic, C Thorne, PA Tovo, G Tudor-Williams, AS Walker, S Welch, U Wintergerst, N Valerius.

**Participants, families, and staff**

We thank all the children, families, and staff from the centers participating in the PENPACT-1 trial. We thank the Children's Mercy Medical Writing Center for copyediting this document.

## Author Contributions

**Conceptualization:** Dwight E. Yin, Stephen R. Cole, Meredith G. Warshaw, Ross E. McKinney, Jr.

**Data curation:** Dwight E. Yin, Christina Ludema, Meredith G. Warshaw.

**Formal analysis:** Dwight E. Yin, Christina Ludema.

**Funding acquisition:** Ross E. McKinney, Jr.

**Investigation:** Ross E. McKinney, Jr.

**Methodology:** Dwight E. Yin, Stephen R. Cole, Carol E. Golin, William C. Miller, Meredith G. Warshaw, Ross E. McKinney, Jr.

**Project administration:** Dwight E. Yin, Stephen R. Cole, Ross E. McKinney, Jr.

**Resources:** Dwight E. Yin.

**Software:** Dwight E. Yin.

**Supervision:** Stephen R. Cole, Ross E. McKinney, Jr.

**Validation:** Dwight E. Yin, Christina Ludema.

**Visualization:** Dwight E. Yin, Christina Ludema.

**Writing – original draft:** Dwight E. Yin.

**Writing – review & editing:** Dwight E. Yin, Christina Ludema, Stephen R. Cole, Carol E. Golin, William C. Miller, Meredith G. Warshaw, Ross E. McKinney, Jr.

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
