## [Decision Letter · Decision Letter 0]

30 Jun 2020

Please carefully address the reviewer's comments and suggestions.

We look forward to receiving your revised manuscript.

Kind regards,

Patricia Evelyn Fast, MD, Ph.D.

Academic Editor

PLOS ONE

Additional Editor Comments:

This paper applies a new analysis to the PENPACT-1 data, to investigate time to treatment interruption for Protease-Inhibitor-based vs NNRTI-based regimens over a 4 year period. The paper is well written and convincing. Two reviewers have provided valuable suggestions to improve the presentation of results.

In addition to addressing the reviewers' comments, please comment on whether important changes in the treatment regimens used, including the availability of palatable protease inhibitors have occurred during the years since the trial ended.

Journal Requirements:

"Although the authors have no financial competing interests for this manuscript, DEY has been an investigator on studies unrelated to HIV supported by Astellas, Chimerix, Merck, Pfizer, Viracor-Eurofins, Kansas City Area Life Sciences Institute, and the Marion Merrell Dow Fund; DEY was a founder and is an unpaid advisory board member for the non-profit Maipelo Children’s Trust/Cover the Globe, which provides HIV services in Botswana. The other authors report no conflicts of interests. "

4. One of the noted authors is a group 'PENPACT-1 (PENTA 9 / PACTG 390) Study Team'. In addition to naming the author group and listing the individual authors and affiliations within this group in the acknowledgments section of your manuscript, please also indicate clearly a lead author for this group along with a contact email address.

Reviewers' comments:

Reviewer's Responses to Questions

**Comments to the Author**

1. Is the manuscript technically sound, and do the data support the conclusions?

Reviewer #1: Yes

Reviewer #2: Yes

2. Has the statistical analysis been performed appropriately and rigorously? 

Reviewer #1: Yes

Reviewer #2: Yes

3. Have the authors made all data underlying the findings in their manuscript fully available?

Reviewer #1: No

Reviewer #2: No

4. Is the manuscript presented in an intelligible fashion and written in standard English?

Reviewer #1: Yes

Reviewer #2: Yes

5. Review Comments to the Author

Reviewer #1: Absence of evidence is not evidence of absence so said the late great Doug Altman writing along with his longtime co-author Martin Bland back in 1985 in their great Statistics Notes series in the BMJ. The caution raised there, I think, applies somewhat to the well presented and succinct piece of work laid out in this mansuscript. The present study is based on an extended followup of a large trial that had an extensive geographical footprint looking at paediatric HIV within children aged 1 month through to just under 18 years. The parent trial was a 2x2 factorial study whose aim was to look at changes in viral load after 4 years. The present study looked to asesss whether tehre was a differenec in the timing of treatment disruption both at the end of the initial followup period which was 4 years and at the end of study at 6.5 years.

I have some issue with some of the wording and presentation of the data in lines 194 through 197. Specifically I would like to see 95% Confidence Intervals for all the estimates given ie the 66% and 83% on line 195, the 70% and 63% on line 197 and the estimates of 81% and 84% on line 212. I believe the term "treatment distruption probability" on lines 195 and 196 would be better put as eg "66% 95CI(LB,UB) had a treatment disruption" and on line 196 consider changing "At 4 years..." to something like "By four years 70% (95 CI(LB,UB) and 63%... had experienced at least one treatment disruption respectively".

Table 2 seems rather unnecessary given that all the information it contains already and fully exists within the text on lines 199-201 and 215-6.

I would suggest that a) given this is secondary analysis of pre-existing data and b) that this hints a little at a post-hoc power analysis, the last sentence of the results could be excised without any loss of import for this work.

Finally, I always find the use of a risk table (showing numbers at risk and events occuring within a time interval) under KM plots to be most helpful and would respectfully request that the authors consider adding these to their Figure 2.

Reviewer #2: The authors have not made the underlying data readily available; access is restricted to protect the privacy and confidentiality of the study participants. The authors have provided an institutional email address through which researchers can access the primary data. This is in line with the Plos data policy.

The authors need to clarify in the ethics statement why exemption from an IRB outside the USA was not sought for the secondary analysis.

Line 76, consider stating that '...children progress much faster to AIDS and death' rather than refer to HIV disease.

Line 78 implies that longer time on their initial regimen means greater efficacy. Please confirm whether there is any evidence that duration of treatment improves efficacy of ART; in the context you have used, it might be more appropriate to state that longer time on the initial regimen may result in greater effectiveness of ART.

Line 116; please confirm if block randomization was carried out with blocks based on the site of enrollment.

Line 124; it would be important to clarify what data was collected on the adherence questionnaires that were administered every 24 weeks and what data was collected during the other scheduled visits, or ad hoc as stated in line 122.

As it reads now, it implies that data on treatment disruptions for each drug were collected covering 3 days prior to every 24 week visit when the questionnaire was administered and barriers to adherence experienced in the 2 weeks prior to this visit. This is a very narrow window to collect this data over 4 years.

Line 146 implies that the adherence questionnaire was administered strictly every 24 weeks. As the visit schedule meant volunteers were seen at least once every 12 weeks, please clarify if it was permissible for an adherence questionnaire to e administered in the visit immediately following a 24 week visit when it was missed.

Lines 167 and 173; please confirm whether the classification of race was based on self-reporting at all sites.

Line 220; please confirm if the 25% of events reported, involved a substitution of either the PI or NNRTI or if these cases include instances of substitution of one or more of the NRTIs without interference with the administration of the PI or NNRTI in the regimen.

Line 221; please clarify if stoppage or suspension of the entire first line regimen resulted in starting the second line regimen in all cases. Please also specify how the group that had the first line regimen stopped differs from the group that had switches to a second-line regimen.

Table 3; Please clarify if caregiver requests were distinguished from adverse events. Please state in line 24 if the caregiver requests listed in the table were all those that were for reasons other than adverse events.

Line 332; please clarify in the methods section if the adherence questionnaire was standardized across study sites and age groups. The statement on line 332 implies that the questionnaires differed across sites and age groups.

Figure 1; what was the reason for volunteers who withdrew consent after initiation of ART?

There were more cases of withdrawal of consent in the NNRTI group than the PI group; please state whether any of these withdrawals were related to adverse events or issues of tolerability of the drugs?

6. PLOS authors have the option to publish the peer review history of their article (what does this mean?). If published, this will include your full peer review and any attached files.

Reviewer #1: **Yes: **Greg Fegan

Reviewer #2: **Yes: **Vincent Muturi-Kioi

---

## [Author Response · Author response to Decision Letter 0]

30 Sep 2020

Thank you for your kind, thoughtful review of our manuscript. We have revised the manuscript and included a letter responding to academic editor and reviewer comments. Our responses are also listed below.

Editor Comments

1. “In addition to addressing the reviewers' comments, please comment on whether important changes in the treatment regimens used, including the availability of palatable protease inhibitors have occurred during the years since the trial ended.”

We address this point on pages 16 and 18 of the revised manuscript.

2. “Please ensure that your manuscript meets PLOS ONE's style requirements, including those for file naming. The PLOS ONE style templates can be found at [Websites removed].”

We have modified our style accordingly.

3. “Thank you for stating the following in the Competing Interests section:

‘Although the authors have no financial competing interests for this manuscript, DEY has been an investigator on studies unrelated to HIV supported by Astellas, Chimerix, Merck, Pfizer, Viracor-Eurofins, Kansas City Area Life Sciences Institute, and the Marion Merrell Dow Fund; DEY was a founder and is an unpaid advisory board member for the non-profit Maipelo Children’s Trust/Cover the Globe, which provides HIV services in Botswana. The other authors report no conflicts of interests.’

Please confirm that this does not alter your adherence to all PLOS ONE policies on sharing data and materials, by including the following statement: "This does not alter our adherence to PLOS ONE policies on sharing data and materials.” (as detailed online in our guide for authors http://journals.plos.org/plosone/s/competing-interests). If there are restrictions on sharing of data and/or materials, please state these. Please note that we cannot proceed with consideration of your article until this information has been declared.”

We confirm that these competing interests do not alter our adherence to PLOS ONE policies on sharing data and materials, and we have included this statement in our resubmission.

4. “Please include your updated Competing Interests statement in your cover letter; we will change the online submission form on your behalf.”

We have included our updated Competing Interests statement in our cover letter. 

5. “We note that you have indicated that data from this study are available upon request. PLOS only allows data to be available upon request if there are legal or ethical restrictions on sharing data publicly. For information on unacceptable data access restrictions, please see http://journals.plos.org/plosone/s/data-availability#loc-unacceptable-data-access-restrictions.

We will update your Data Availability statement on your behalf to reflect the information you provide.”

IMPAACT and PENTA are willing to share the data, upon review and approval of a data request. The information for such a data request are provided.

Please note that the original data were obtained by our own data request to IMPAACT, and publication of our results required approval of PACTG/IMPAACT and PENTA. As such, the data are not our own to share, but rather are of PACTG/IMPAACT and PENTA. In addition, the data we received are both potentially identifiable, as determined by the multiple IRBs listed in the manuscript. Moreover, the data are considered sensitive, as they include the HIV diagnosis of participants in the trial. Therefore, receipt of the data for our analysis was contingent upon our not sharing the data, given its sensitive nature. Our original statement on not being able to share the data is verbatim from the IMPAACT instructions we were given. 

6. “One of the noted authors is a group 'PENPACT-1 (PENTA 9 / PACTG 390) Study Team'. In addition to naming the author group and listing the individual authors and affiliations within this group in the acknowledgments section of your manuscript, please also indicate clearly a lead author for this group along with a contact email address.”

The listed senior author of this manuscript is Ross E. McKinney, Jr., who was a North American co-chair of the study and one of the authors of the parent study primary manuscript. We have included his contact email address on page 2.

5. “Please include captions for your Supporting Information files at the end of your manuscript, and update any in-text citations to match accordingly. Please see our Supporting Information guidelines for more information: http://journals.plos.org/plosone/s/supporting-information.”

We have included captions for the Supporting Information files and updated in-text citations accordingly. 

Reviewer comments

Reviewer 1

1. “Absence of evidence is not evidence of absence so said the late great Doug Altman writing along with his longtime co-author Martin Bland back in 1985 in their great Statistics Notes series in the BMJ. The caution raised there, I think, applies somewhat to the well presented and succinct piece of work laid out in this manuscript. The present study is based on an extended followup of a large trial that had an extensive geographical footprint looking at paediatric HIV within children aged 1 month through to just under 18 years. The parent trial was a 2x2 factorial study whose aim was to look at changes in viral load after 4 years. The present study looked to assess whether there was a difference in the timing of treatment disruption both at the end of the initial followup period which was 4 years and at the end of study at 6.5 years.”

Thank you for your comments. We agree that absence of evidence is not evidence of absence. We have re-worded our statements to reflect that our estimates are not compatible with large differences between treatment groups in time to treatment disruption. Changes are noted on page 15.

2. “I have some issue with some of the wording and presentation of the data in lines 194 through 197. Specifically I would like to see 95% Confidence Intervals for all the estimates given ie the 66% and 83% on line 195, the 70% and 63% on line 197 and the estimates of 81% and 84% on line 212. I believe the term "treatment disruption probability" on lines 195 and 196 would be better put as eg "66% 95CI(LB,UB) had a treatment disruption" and on line 196 consider changing "At 4 years..." to something like "By four years 70% (95 CI(LB,UB) and 63%... had experienced at least one treatment disruption respectively".”

We agree with including 95% confidence intervals with these estimates and have inserted them as recommended on pages 12-13.

2. “Table 2 seems rather unnecessary given that all the information it contains already and fully exists within the text on lines 199-201 and 215-6.”

We appreciate Reviewer #1’s suggestion and have eliminated this table. We have renumbered our tables accordingly.

3. “I would suggest that a) given this is secondary analysis of pre-existing data and b) that this hints a little at a post-hoc power analysis, the last sentence of the results could be excised without any loss of import for this work.”

Thank you for pointing out that this information suggests a post-hoc power analysis. This information was included as a sensitivity analysis to show potential influences of various treatments of missing questionnaire data. However, given the potential for misinterpretation, we have removed this sentence (page 15) and removed the corresponding descriptions from the Methods (page 9).

4. “Finally, I always find the use of a risk table (showing numbers at risk and events occuring within a time interval) under KM plots to be most helpful and would respectfully request that the authors consider adding these to their Figure 2.”

We appreciate this suggestion and have revised Figure 2 to include the numbers at risk, censored, and events occurring within the time intervals.

Reviewer 2

1. “The authors have not made the underlying data readily available; access is restricted to protect the privacy and confidentiality of the study participants. The authors have provided an institutional email address through which researchers can access the primary data. This is in line with the Plos data policy.”

Thank you for discussing this matter. As per academic editor recommendations, we have submitted the privacy risks and sensitive nature of the data, as well as recognizing that the data are not ours to share, but rather were obtained by our own data request. We have included the official statement of IMPAACT regarding data access and the mechanism by which these data may be requested.

2. “The authors need to clarify in the ethics statement why exemption from an IRB outside the USA was not sought for the secondary analysis.”

As these data were received via a data request from the central data management center, we only sought approval from the institutional directly overseeing the investigators performing the secondary analysis. We have included this clarification in the Methods section (page 7).

3. “Line 76, consider stating that '...children progress much faster to AIDS and death' rather than refer to HIV disease.”

Thank you for this important distinction. We have edited this sentence accordingly (page 5).

4. “Line 78 implies that longer time on their initial regimen means greater efficacy. Please confirm whether there is any evidence that duration of treatment improves efficacy of ART; in the context you have used, it might be more appropriate to state that longer time on the initial regimen may result in greater effectiveness of ART.”

Thank you for this clarification. We have re-phrased as per the reviewer’s suggestion, cited data showing that 2nd line treatment failure is common and problematic on page 5, and cited corroborating data on pediatric first-line durability on page 17. 

5. “Line 116; please confirm if block randomization was carried out with blocks based on the site of enrollment.”

Randomization was both stratified and in blocks. Stratification was within categories of age, receipt of perinatal ART, and within the research network (PACTG or PENTA), not by site. Trial statisticians employed variable block sizes. We have clarified the blocked randomization design in the manuscript (page 7).

6. “Line 124; it would be important to clarify what data was collected on the adherence questionnaires that were administered every 24 weeks and what data was collected during the other scheduled visits, or ad hoc as stated in line 122. As it reads now, it implies that data on treatment disruptions for each drug were collected covering 3 days prior to every 24 week visit when the questionnaire was administered and barriers to adherence experienced in the 2 weeks prior to this visit. This is a very narrow window to collect this data over 4 years.”

We have clarified the more frequent clinic visits during which the treatment records were recorded and the less frequent, 24-weekly visits on which the adherence questionnaire were collected (pages 7-8). We combined measures to balance the narrow window of data collection identified by the reviewer. 

7. “Line 146 implies that the adherence questionnaire was administered strictly every 24 weeks. As the visit schedule meant volunteers were seen at least once every 12 weeks, please clarify if it was permissible for an adherence questionnaire to e administered in the visit immediately following a 24 week visit when it was missed.”

Yes, if the visitor adherence questionnaire was missed at 24 weekly intervals, the adherence questionnaires may have been, and often were, administered at the following visit. We have included this information in the manuscript (page 7).

8. “Lines 167 and 173; please confirm whether the classification of race was based on self-reporting at all sites.”

Yes, race/ethnicity classification was based on self-report or report of the parent/caregiver.

9. “Line 220; please confirm if the 25% of events reported, involved a substitution of either the PI or NNRTI or if these cases include instances of substitution of one or more of the NRTIs without interference with the administration of the PI or NNRTI in the regimen.”

Due to the lack of granularity in the adherence questionnaire, we could not distinguish whether questionnaire-reported treatment disruptions were only due to PI or NNRTI vs. NRTI backbone with confidence. Such could be distinguished on the treatment record. Thus, we performed the primary analysis using the least common denominator, which was disruption of any component of the initial ART regimen (page 8, 19). To account for potential influence of changing only the NRTI backbone, we also performed sensitivity analyses including only changes in the PI or NNRTI, as per the treatment records, recognizing that we would be mis-specifying events from the adherence questionnaire (page 9). The results were largely the same as the primary analysis.

10. “Line 221; please clarify if stoppage or suspension of the entire first line regimen resulted in starting the second line regimen in all cases. Please also specify how the group that had the first line regimen stopped differs from the group that had switches to a second-line regimen.”

A stoppage or suspension of the entire first line regimen may have led to either a permanent discontinuation or a subsequent switch to a 2nd line regimen (page 8, line 142), and we explored different definitions for duration of time off therapy that qualified as a treatment discontinuation (pages 9, line 160; 15, lines 264-266). Permanent discontinuations would also include such events as loss-to-follow-up or withdrawal from study. As a result, we do not have measurement of potential future re-starting of ART for patients where were lost or withdrew. 

11. “Table 3; Please clarify if caregiver requests were distinguished from adverse events. Please state in line 24 if the caregiver requests listed in the table were all those that were for reasons other than adverse events.”

The treatment record allowed listing of only one reason for stops or changes in the initial treatment regimen. Although subsequent information from future visits may suggest that some caregiver requests were related to adverse events, differentiating caregiver requests that were or were not related to adverse events requires the authors to read into the records, rather than reporting on the records. Thus, we chose to use the reason first listed at the treatment disruption event, which we expect would introduce less author bias in interpretation. We have added a comment in the Methods on page 8 to clarify. 

12. “Line 332; please clarify in the methods section if the adherence questionnaire was standardized across study sites and age groups. The statement on line 332 implies that the questionnaires differed across sites and age groups.”

Thank you for this comment. We now address this issue on page 7 of Methods. Each research network (PENTA or PACTG) had its own standardized questionnaires. Key information was harmonized across questionnaires, but each network used their own standardized questionnaire with some remaining differences.

We are willing to provide copies of the adherence questionnaires as Supplementary Material if the Editor deems this necessary.

13?. “Figure 1; what was the reason for volunteers who withdrew consent after initiation of ART? There were more cases of withdrawal of consent in the NNRTI group than the PI group; please state whether any of these withdrawals were related to adverse events or issues of tolerability of the drugs?”

Unfortunately, we do not have access to these reasons for withdrawal of consent.

---

## [Decision Letter · Decision Letter 1]

3 Nov 2020

Time to treatment disruption in children with HIV-1 randomized to initial antiretroviral therapy with protease inhibitors versus non-nucleoside reverse transcriptase inhibitors

PONE-D-20-04892R1

Dear Dr. Yin,

We’re pleased to inform you that your manuscript has been judged scientifically suitable for publication and will be formally accepted for publication once it meets all outstanding technical requirements.

Kind regards,

Patricia Evelyn Fast, MD, Ph.D.

Academic Editor

PLOS ONE

Additional Editor Comments (optional):

Thank you for your careful attention to the reviewers' comments.

Reviewers' comments:

Reviewer's Responses to Questions

**Comments to the Author**

1. If the authors have adequately addressed your comments raised in a previous round of review and you feel that this manuscript is now acceptable for publication, you may indicate that here to bypass the “Comments to the Author” section, enter your conflict of interest statement in the “Confidential to Editor” section, and submit your "Accept" recommendation.

Reviewer #1: All comments have been addressed

Reviewer #2: All comments have been addressed

2. Is the manuscript technically sound, and do the data support the conclusions?

Reviewer #1: (No Response)

Reviewer #2: Yes

3. Has the statistical analysis been performed appropriately and rigorously? 

Reviewer #1: (No Response)

Reviewer #2: Yes

4. Have the authors made all data underlying the findings in their manuscript fully available?

Reviewer #1: (No Response)

Reviewer #2: No

5. Is the manuscript presented in an intelligible fashion and written in standard English?

Reviewer #1: (No Response)

Reviewer #2: Yes

6. Review Comments to the Author

Reviewer #1: (No Response)

Reviewer #2: (No Response)

7. PLOS authors have the option to publish the peer review history of their article (what does this mean?). If published, this will include your full peer review and any attached files.

Reviewer #1: **Yes: **Greg Fegan

Reviewer #2: **Yes: **Vincent Muturi-Kioi

---

## [Editor Report · Acceptance letter]

12 Nov 2020

PONE-D-20-04892R1 

Time to treatment disruption in children with HIV-1 randomized to initial antiretroviral therapy with protease inhibitors versus non-nucleoside reverse transcriptase inhibitors 

Dear Dr. Yin:

I'm pleased to inform you that your manuscript has been deemed suitable for publication in PLOS ONE. Congratulations! Your manuscript is now with our production department. 

Kind regards, 

on behalf of

Dr. Patricia Evelyn Fast 

Academic Editor

PLOS ONE